# Behavior Selection Metaheuristic Search Algorithm for the Pollination Optimization: A Simulation Case of Cocoa Flowers

**Willa Ariela Syafruddin \*** , **Rio Mukhtarom Paweroi and Mario Köppen**

Department of Computer Science and System Engineering (CSSE), Graduate School of Computer Science and System Engineering, Kyushu Institute of Technology, 680-4 Kawazu, Fukuoka 820-8502, Japan; paweroi.rio-mukhtarom223@mail.kyutech.jp (R.M.P.); mkoeppen@ieee.org (M.K.)
**\*** Correspondence: syafruddin.willa-ariela966@mail.kyutech.jp

**Abstract:** Since nature is an excellent source of inspiration for optimization methods, many optimization algorithms have been proposed, are inspired by nature, and are modified to solve various optimization problems. This paper uses metaheuristics in a new field inspired by nature; more precisely, we use pollination optimization in cocoa plants. The cocoa plant was chosen as the object since its flower type differs from other kinds of flowers, for example, by using cross-pollination. This complex relationship between plants and pollinators also renders pollination a real-world problem for chocolate production. Therefore, this study first identified the underlying optimization problem as a deferred fitness problem, where the quality of a potential solution cannot be immediately determined. Then, the study investigates how metaheuristic algorithms derived from three well-known techniques perform when applied to the flower pollination problem. The three techniques examined here are Swarm Intelligence Algorithms, Individual Random Search, and Multi-Agent Systems search. We then compare the behavior of these various search methods based on the results of pollination simulations. The criteria are the number of pollinated flowers for the trees and the amount and fairness of nectar pickup for the pollinator. Our results show that Multi-Agent System performs notably better than other methods. The result of this study are insights into the co-evolution of behaviors for the collaborative pollination task. We also foresee that this investigation can also help farmers increase chocolate production by developing methods to attract and promote pollinators.

**Keywords:** metaheuristic search algorithm; swarm intelligence; random search; multi-agent systems; optimization; behavior; pollination of cocoa flowers





## 1. Introduction

Pollination is a natural process that is required for most plants to produce fruit and seeds. Cocoa is one of the plants that depend on pollination for successful fruit formation [1]. The cacao plant (*Theobroma cacao*) grows almost everywhere globally, but it is most common in tropical areas such as West Africa, Indonesia, Central and South America, and Hawaii. This area issue creates a concern for chocolate production because cocoa flower pollinators prefer humid environments. Only a few fly genera, especially a tiny midge called *Forcipomyia Inornatipennis* (FP), can pollinate small flowers. The problem is because cacao flowers are unlike other flowers. Since the cacao flower is small and almost odorless, it does not attract the attention of many insects, especially not classical pollinators such as bees. FP wander to cacao flowers in order to obtain the nectar in bloom for food and egg maturation. The pollination process in cacao flowers can be summarized as follows: when the FP takes the nectar from the cacao flower, the FP inadvertently touches the bunch's head, causing pollen to be released and to stick to the FP. Pollination will occur by chance when the pollen picked up by FP on one flower meets the pollen on another cacao flower that the FP flies to.

Self-incompatible tree species are primarily cross-compatible, which means they can fertilize flowers on other trees, including those of the same variety. Cross-pollination is the

only method to ensure successful fertilization because self-pollination is not suitable for cacao trees and will not result in successful fertilization [2]. This situation implies matching behaviors of the pollinators. We aim to investigate this process by pollination simulations on cocoa plants by using three methods of pollinator collaboration:

- Based on Swarm Intelligence Algorithms;
- Based on Individual Random Search;
- Based on Multi-Agent Systems differential search methods.

The study of optimization algorithm behavior to tackle pressing real-world problems has recently attracted many researchers' attention. Currently, new algorithms are set up and focused on achieving a pre-set desired optimization goal. While this can be useful and efficient in the short term, it is insufficient in the long run as it needs to be repeated for any new problem that occurs under potentially new specific difficulties. Therefore, one algorithm cannot be used for all real-world issues.

The development of optimization algorithms is essential because optimization problems can occur in various scientific fields such as economics, engineering, and medicine. There are still many researchers around the world working to solve problems in this field. Classical optimization algorithms are not very efficient at solving real-world problems because they cannot find the global optima or it requires massive efforts. Bn contrast, metaheuristic algorithms are more robust at avoiding local optima. They do not need a cost function gradient because the algorithm's main "trick" is also to exploit randomness and concurrency [3].

The paper focuses on theoretical and empirical research investigating approaches needed to analyze stochastic optimization algorithms and performance assessment concerning different criteria. Figure 1 shows the FP process to pollinate cocoa plants. The FP can use different search methods to ensure that the FP can collect enough nectar. However, if the tree pollination has occurred, the tree will no longer expose flowers and not produce nectar. Consequently, FP will have to seek out other trees that have not yet pollinated to obtain more nectar. This processing is then implemented in a 3D virtual environment to demonstrate and compare the effectiveness of the various cocoa pollination methods.

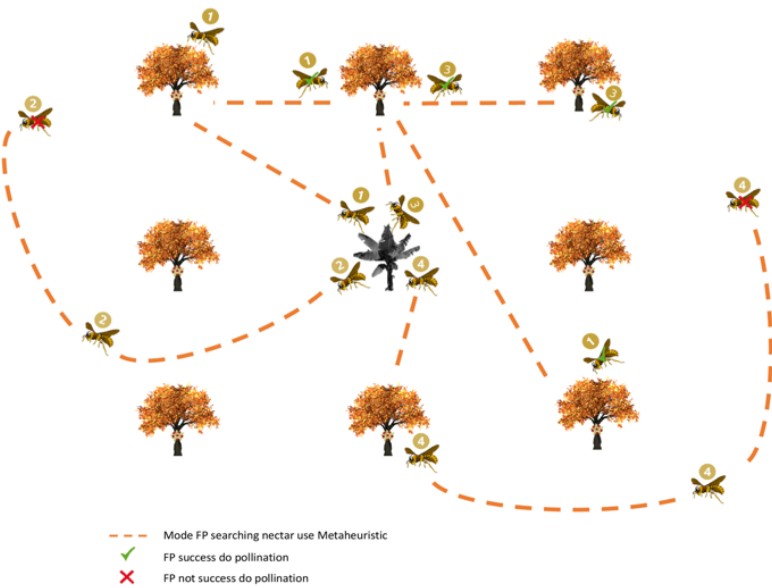

- - - 　Mode FP searching nectar use Metaheuristic
✓　　FP success do pollination
✗　　FP not success do pollination

**Figure 1.** Process flow of *Forcipomyia* pollination.

This article is organized as follows. In Section 2, the real-world facts on cocoa pollination are introduced. Section 3 provides the material and methods used in this study. Results are presented in Section 4, followed by discussions (Section 5) and conclusions (Section 6).

## 2. Real-World Pollination Optimization Problem

Cocoa flower pollination uses a different method of pollination compared to flower pollination in general. The flowers on these cacao plants are distinct from other plants since they are small (typical diameter of about 3 cm), which allows the flowers to be pollinated by only small-bodied insects (most excluding traditional pollinators such as bees). The flower's structure with the characteristic hooded petals enclosing the stamens favors neither self-pollination nor cross-pollination. However, there is considerable evidence from various sources that natural crossing occurs in a substantial amount. The reason is that the arrangement of cocoa flowers has distinctive veil petals which protect stamens that do neither prefer self-pollination nor cross-pollination, even though there are clear indications from several sources that natural cross-pollination occurs to a certain extent [4]. Based on the findings of Jones' (1912) experiments in Dominica, it is clear that small insects, whether ants, aphids, thrips, or a combination of the three, are the primary agents involved in the pollination of cocoa flowers [5].

*Forcipomyia inornatipennis* swarms can be classified into (1) normal and (2) mating swarms: [6]:

- Normal swarm: The standard FP flight has a 12 h cycle. Swarming activity affecting the highest number of insects occurs between 5 a.m. and 8 a.m., during which it rapidly declines to its lowest level, from about midday to around 2 p.m., where it starts to rise to a second high between 5 a.m. and 6:30 a.m. However, variation in the behavior of the insects depends on the light of the sun. If the sky is overcast, the dawn swarm will begin until after 9 a.m. A swarm that is 30–180 cm above the ground and consists of 4–80 individuals of both sexes may fly in either direction within a 100 cm radius. The higher the swarm, the more midges leave; thus, the remaining number of individuals is directly proportional to those engaged in the hive.
- Mating swarm: Almost 60% of mating swarm activities occur at dusk. The swarm has 2 to 30 individuals depending on the time of day and both hives have both sexes. In flight, males actively hunt for females and male mates in a swarm usually last around 15 min with three or four females. This habit is because the number of females taking part in flights is always smaller than males. If the swarm does not contain more than four or five pairs, the number of remaining midges will be determined by the mating swarm's size, as long as it exists.

Cocoa flowers have a different shapes compared to flowers in general, rendering them unattractive to specific potential pollinators. Cocoa flowers have different staminodes, as shown in Figure 2. These staminodes are similar to stamens, which is the male part of the flower, but they do not contain pollen and, thus, they are sterile. Only tiny insects at the flower's base can reach the pollen-producing anthers hidden beneath a hood. The research conducted by Frimpong-Anin [7] claimed that the variants of converging and parallel staminode flowers were the ideal types of staminode-style flowers for successful pollination.

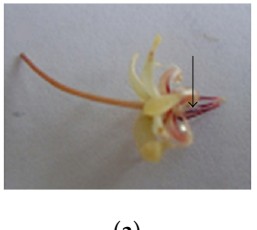 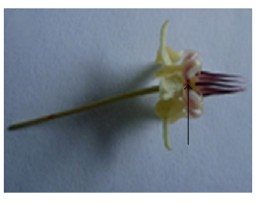 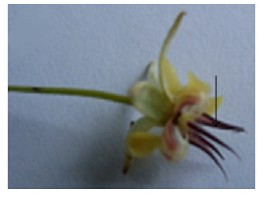

(**a**)　　　　　　　　　　　　　　　　(**b**)　　　　　　　　　　　　　　　　(**c**)

**Figure 2.** Three variants of the Cocoa Flowers. (**a**) Converging. (**b**) Parallel. (**c**) Splay.

Flower pollination is a fascinating natural process that has captivated several authors who have researched it. The target of flower pollination is to ensure the survival of the most suitable and optimal plant reproduction in terms of both number and quality. All of the flower pollination factors and processes mentioned above interact to ensure that flowering

plants reproduce optimally. Therefore, the author [8] introduced a new algorithm inspired by the perspective of flower pollination. The authors [9] developed a pollination simulation model and the results revealed that seed production is affected by pollinator and pollen carrier movement patterns. Another author [10] was inspired by the relationship between pollinating insects and flowering plants and presented an agent-based simulation to assess the potential impact of heterospecific pollen transfer by insects on two species of flowering plants in an environment that included a shared central region and specific-species refugia.

From the above description, the FP system for pollinating the flowers of the cocoa plant differs from the characteristics of the flowers in general. Therefore, three methods are proposed and then evaluated against the FP search method to observe how each method behaves. Many researchers have shown that eacg living creature in this world possess different selection behavior [11]. Based on this conclusion, a metaheuristic algorithm is proposed to assist FP in space exploration. The same can be said about another metaheuristic algorithm significantly inspired by animal behavior [12–17].

We have adopted the FP's pollination method based on ideas and concepts introduced in [6]. This process was implemented by using a simulation. Figure 3 shows the flowchart of our proposed methodology. We create an environment that follows the original scenario, such as a cocoa tree with tiny flowers with FP insects. As mentioned earlier, the FP and tree have a complicated relationship because the tree cannot self-pollinate and the FP only wants nectar from the cacao flower. Therefore, FP and tree concurrently follow different flowcharts. Figure 3a shows how the FP is looking for nectar, while Figure 3b shows how trees can be pollinated by FPs unintentionally by bringing pollen attached to their body from different trees. Pollinating conditions in this image indicate that pollination will occur when an is FP closer to the non-pollinated tree, the nectar amount of the FP increases, and the FP carries poll from that tree. If the FP already carries poll from another tree, the tree becomes pollinated.

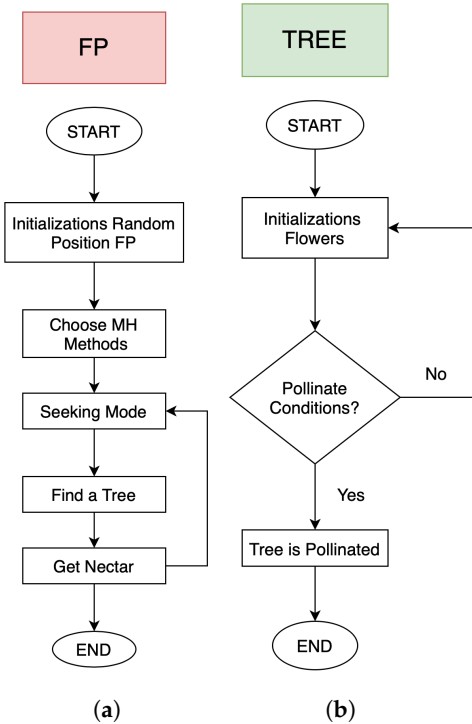

(**a**)                                  (**b**)

**Figure 3.** Flowchart of the pollination model used here: (**a**) FP. (**b**) Tree.

## 3. Material and Methods

### 3.1. Material

For the simulation, a cloud-hosted instance of the OpenSimulator server (version 0.9.1) was used. This OpenSimulator provides a suitable environment for performing the study

for offering various frameworks such as server-client architecture, grid architecture, avatar-based control, concurrency, and scripting support. Within the so-called hypergrid linking the different server simulations worldwide, it became possible to design an experimental framework for conducting simulations that can be tested, analyzed, and upgraded through multi-institutional collaboration [18].

All the experiments for this study were performed on Dual-Core Intel Core i5 Mac-Book @ 3.1 GHz with 8 GB 2133 MHz LPDDR3 of RAM running the viewer (client) software FirestormOS-Releasex64. The OpenSimulator server was running in the Metropolis Metaversum grid hosted by Hypergrid Virtual Solution UG, as illustrated in Figure 4.

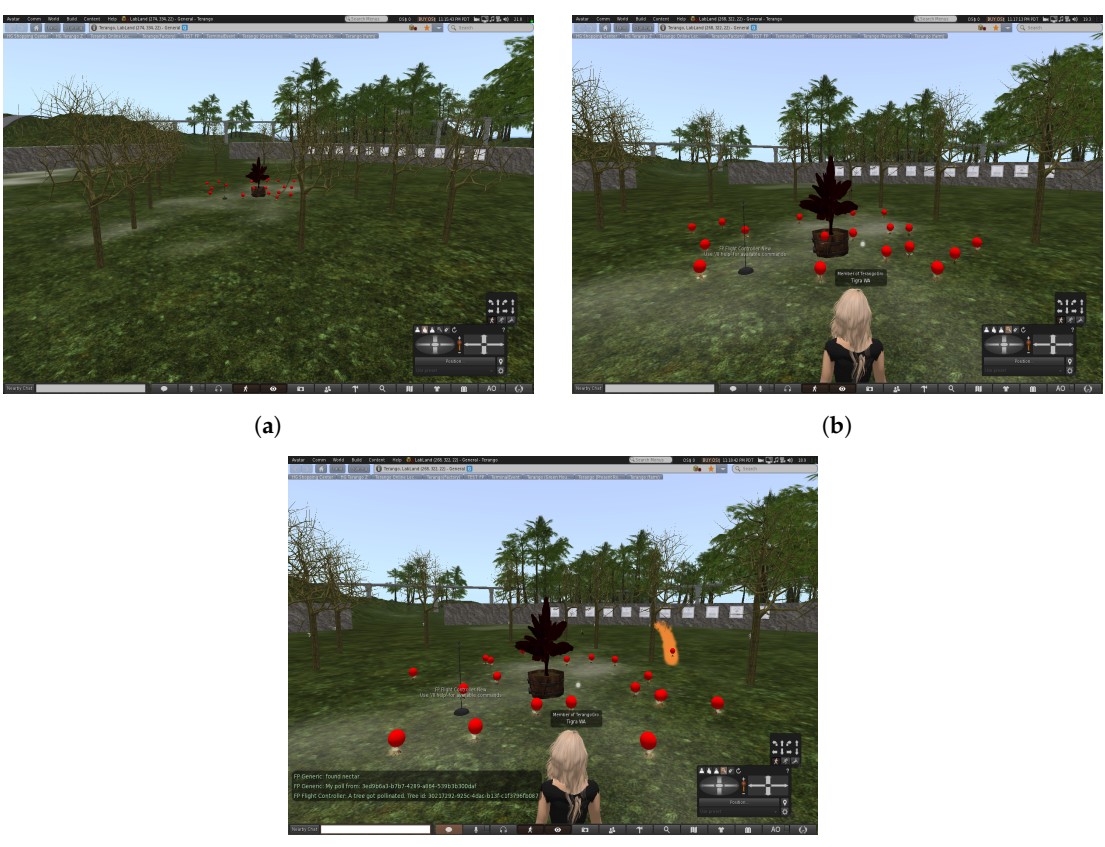

**Figure 4.** Implemented simulation in a 3D virtual environment. (**a**) FP randomly gathering around the breeding site. (**b**) FP starts searching trees. (**c**) A tree becomes pollinated.

Experimental Environment

The purpose of the experiment was to employ simulated creatures that comply with FP pollinating cocoa plant flowers by considering the use of the metaheuristics search algorithm. Utilizing various metaheuristic searches in this research study allows an almost identical real-life scenario, rendering it easy to observe surprising outcomes and to explore the benefits or drawbacks of each metaheuristic search used from different perspectives. In order to accomplish this goal and to calculate the efficiency of the studied metaheuristic search, the following scenarios are considered:

- FP foraging usually starts from dark moist places such as rotting banana trees, which is also the breeding site for FP;
- The starting point of the FP is random but is nearby its breeding site;
- Then, the follow three scenarios for a fixed period can occur: the first case where the number of reachable trees and the number of FP become the same; the second case where there are more reachable trees than FP; and the third case is where there are more FPs than reachable trees.

**Table 1.** Scenario of experiment.

|  | Experiment 1 | Experiment 2 | Experiment 3 |
|---|---|---|---|
| FP | 10 | 10 | 20 |
| Tree | 10 | 15 | 15 |
| Time (minutes) | 20 | 20 | 20 |
| Number of Simulations | 20 | 20 | 20 |

Table 1 and Figure 5 describe how the simulation works in each experiment. For additional information, the following explains how the simulation works in detail:

- The simulation occurs in circle space with a diameter of 90 m consisting of FP, tree, and FP breeding sites in the center of the circle space.
- The position of each tree is given in the Figure 5. Since each experiment had various trees, every tree was around 6 to 10 m apart.
- Before the simulation begins, the FP will be positioned at a random location within 5 m of the breeding site and will remain at the same height during the simulation, neither rising nor lowering.
- When the simulation begins, one of the algorithm methods is selected, and FP will start looking for trees in circle space for 20 min without crossing the boundary, with time steps of 1 s for Idle-Jaya, Idle-Cuckoo, Lévy, and DAG and 2 s for Idle-CSA.
- When an FP is closer than 3 m to a non-pollinated tree, the nectar amount of the FP increases and the FP carries poll from that tree. If the FP already carries poll from another tree, the tree become pollinated.
- Each tree exposes one flower. When the FP is near a tree, it will collect one nectar. The flower will then replicate nectar after 30 s and will not produce flowers again if pollination is successful.
- FP will fly around search for the trees till a time limit.

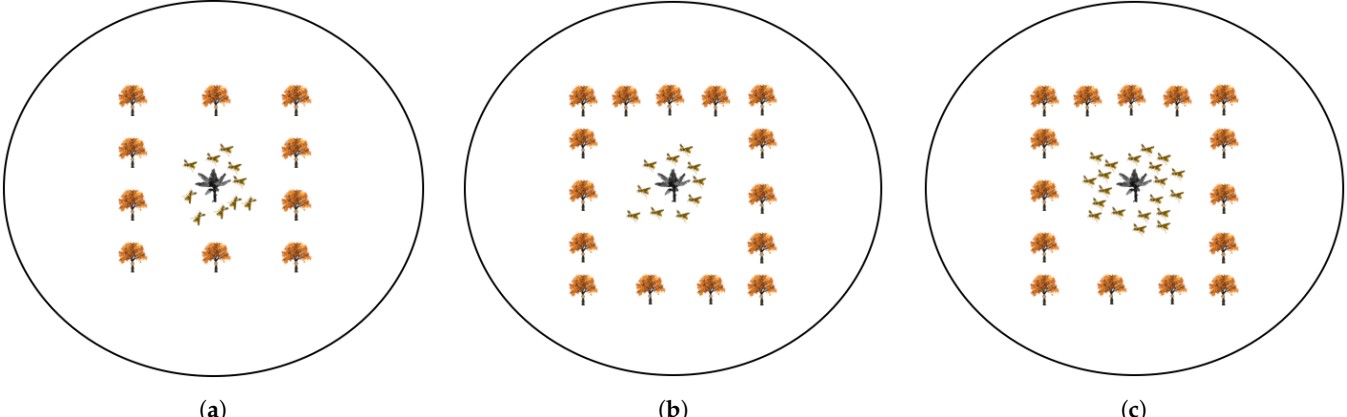

(**a**)          (**b**)          (**c**)

**Figure 5.** Search space environment of experiment. (**a**) Experiment 1. (**b**) Experiment 2. (**c**) Experiment 3.

### 3.2. Methodology

We use three different search methods: methods derived from Swarm Intelligence Algorithms, especially the Jaya algorithm [19], Crow Search Algorithm [20], and Cuckoo Search Algorithm [14]; the second method is the Individual Random Search method, Lévy flight [21]; and the last method is the Multi-Agent Systems differential search, i.e., Defender Aggressor Game (DAG). However, we did not use the original algorithms but had to modify them to fit deferred fitness. This means that we cannot use direct fitness evaluations, as they are needed especially for the Swarm Intelligence Algorithms. The FP also could not focus alone on maximizing nectar intake since this has no direct impact on the number of pollinated trees nor can the trees provide any means to maximize the number of pollinated trees themselves—they still require the FP for that to be possible. However, the Swarm

Intelligence Algorithms can be redesigned to keep only their exploration components and so we used them in "idle-mode" for the exploration part only; fitness value evaluations were not utilized. The algorithm modification is for replacing all fitness-value based internal processing of those algorithms with random decisions. We will introduce it in the following subsection in more detail.

### 3.2.1. Swarm Intelligence Algorithm

This paper proposes three swarm algorithms that can solve the FP pollination problem. Such algorithms, which were inspired by the behavior of social insect colonies and other animal societies, are known as Swarm Intelligence (SI) and are one of the most widely used techniques by researchers to solve complex problems [22–24]. SI, in particular, is frequently used as an inspiration in natural biological systems, involving the collaborative study of the behavior of individuals from populations interacting with one another locally. However, as already explained, we do not use direct fitness evaluations on SI but, instead, a "idle-mode" variant.

Note that the algorithms operate in concurrent modes, each FP described is calculated at a new position on a periodic schedule within the simulation. There is no looping through all FP individuals or any central control of the algorithm. Moreover, generally all FP repositionings are clamped by not exceeding a maximum distance from the hives. A step *Pollination* in the following pseudo-codes is, according to the above descriptions, a tree is pollinated under the condition that current FP already carries poll picked up at a different tree in the preceding algorithm steps.

**Method 1:** Idle-Jaya

The Jaya Algorithm is one of the well-known algorithms used by researchers to solve complex optimization problems. Since this algorithm has no parameters to set, it is known as a simple algorithm. This Algorithm 1 concept is about the same as Particle Swarm Optimization (PSO) [25], but Jaya Algorithm tends toward the best and away from the worst rather than heading to the personal and global best. We used the Idle-Jaya formula in the pseudo-code below and by changing the concept mildly, we moved randomly towards the farther FP and away from the closer FP, which replaces the notion of the best and worst individual in the standard Jaya update formula:

$$X'_{j,k,l} = X_{j,k,l} + r_{1,j,i}\left(X_{j,best,i} - |X_{j,k,l}|\right) - r_{2,j,i}\left(X_{j,worst,i} - |X_{j,k,l}|\right) \tag{1}$$

Notation:

$X_{j,best,i}$　　　　The value of the variable $j$ for the best candidate;
$X_{j,worst,i}$　　　The value of the variable $j$ for the worst candidate;
$X'_{j,k,l}$　　　　　The updated value of $X_{j,k,i}$;
$r_{1,j,i}$ and $r_{2,j,i}$　Random numbers i.i.d. from [0.1].

In this modification dubbed "Idle Jaya", two other FP are randomly selected and the closer one is taken as "worst" and the distant one "best".

---

**Algorithm 1** The Idle-Jaya.

---

1:  Initialize each FP position
2:  Initialize each Tree position
3:  **while** termination condition not satisfied:
4:      $FP_{current}$ choose two another random FP; $FP_1$ and $FP_2$.
5:      Repeat until $FP_1 \neq FP_2$.
6:      Get position of FP;
        $X_{current} = FP_{current}$ position, $X_1 = FP_1$ position, $X_2 = FP_2$ position.
7:      Calculate distance FP; $d_1 = d(X_{current} - X_1)$, $d_2 = d(X_{current} - X_2)$
8:      **if** $d_1 > d_2$:
9:          $X_{far} = X_1$; $X_{near} = X_2$.
10:     **else**
11:         $X_{far} = X_2$; $X_{near} = X_1$.
12:     **end if**
13:     **if** $r > 0.5$ (chance to move to new position):
14:         Normalized vector; $X_{towards} = X_{far} - X_{current}$.
15:         Normalized vector; $X_{away} = X_{near} - X_{current}$.
16:         Calculate new position using; $X_{new} = X_{current} + (r * X_{towards}) - (r * X_{away})$.
17:         Update new position.
18:     **end if**
19:     Calculate distance between FP and nearest Tree; $d_{tree} = d(X_{new} - X_{tree})$.
20:     **if** $d_{tree} \leq 3$ ($T_{nearest}$ found):
21:         **if** FP carry poll of $T_{poll} \neq T_{nearest}$:
22:             Pollination.
23:         **end if**
24:         FP carry poll of $T_{nearest}$; $T_{poll} = T_{nearest}$
25:     **end if**
26: **end while**

---

**Method 2:** Idle-CSA

Crow Search Algorithm (CSA) is a SI algorithm derived from the crow method to store food in a hiding place and to retrieve it when needed. The crow is considered an intelligent bird and Askarzadeh [20] has Stated that a foraging crow resembles an optimization process. In CSA, the concept of deception is incorporated in a SI algorithm. According to the pseudo-code in the Idle-CSA Algorithm 2, one FP can operate as both a position giver and a position receiver at the same time. The per iteration update formula in CSA is as follows.

$$X^{i,iter+1} = X^{i,iter} + r_i x f l^{i,iter} \left( m^{j,iter} - X^{i,iter} \right) \tag{2}$$

Notation:

| | |
|---|---|
| $X^{i,iter}$ | Position of crow $i$ at time *iter* in search space; |
| $r_i$ | Random number with uniform distribution between 0 and 1; |
| $f l^{i,iter}$ | Denotes the flight length of crow $i$ at iteration *iter*; |
| $m^{j,iter}$ | Denotes either the position of hiding place of crow $j$ at time *iter* or a random new location in search space (crows' deception). |

---

**Algorithm 2** The Idle-CSA.

---

 1: Initialize each FP position
 2: Initialize each Tree position
 3: Set reach of step; *reach*.
 4: Set base position; *basepos*.
 5: Set radius; *radius*.
 6: **while** termination condition not satisfied:
 7:     Choose another FP randomly.
 8:     Get a response from that FP.
 9:     *response* = *getResponse*().
10:     <case: receive position>.
11:     Define target move; $m = response$
12:     Get current FP position; $X_{current}$.
13:     Calculate distance between target move and current FP position; $d$.
14:     Calculate new position; $X_{new} = X_{current} + (m - X_{current}) * r * d$
15:     Memorize position; $X_{memory} = X_{new}$
16:     Update new position.
17:     Calculate distance between FP and nearest Tree; $d_{tree} = d(X_{new} - X_{tree})$.
18:     **if** $d_{tree} \leq 3$ ($T_{nearest}$ found):
19:         **if** FP carry poll of $T_{poll} \neq T_{nearest}$:
20:             Pollination.
21:         **end if**
22:     **end if**
23:     **end if**
24:     <case: give position>
25:     *getResponse*():
26:         $X_{memory} = X_{current}$.
27:         **if** $r > awareness$.
28:             *response* = $X_{memory}$.
29:         **else**
30:             *response* = $basepos + r - radius$.
31:         **end if**
32:     *end getResponse*
33: **end while**

---

**Method 3:** Idle-Cuckoo Search via Lévy Flights Algorithm

Cuckoo search is an algorithm that combines the breeding activity of a certain cuckoo species with Lévy flying behavior. Cuckoo search has two search modes: local search and global search, regulated by the redirect probability. As a result, the search space may be examined more effectively globally, increasing the probability of discovering the global optimum. This is because local searches use around one-quarter of the overall search time, whereas global searches use three-quarter of the total search time [26]. As an SI algorithm, the main difference compared to other SI algorithms is that a FP applies the position update of a different individual to itself instead of the other individual, resembling the parasitic habit of a cuckoo putting its eggs into another bird's nest.

$$X_i^{(t+1)} = X_j^t + \alpha \bigoplus Lévy(\lambda) \tag{3}$$

Notation:

$X_i^{(t+1)}$     Generating new solutions $X^{(t+1)}$ for a cuckoo *j*; Lévy step is added
           to position of individual *j*;
$\alpha > 0$     The step size which should be related to the problem scale;
$\alpha$        Weight factor of Lévy step;
$\bigoplus$     Entry-wise multiplications.

Furthermore, cuckoo search is more efficient since the global search uses Lévy flights rather than the typical random walk. In Algorithm 3, we used the recommended value of 0.1 for step size $\alpha$ to avoid too distant moves of FP. The details of the Lévy flight will be discussed in the following algorithm, as this uses the same concept.

---

**Algorithm 3** The Idle-Cuckoo Search via Lévy Flights Algorithm.

---

 1: Initialize each FP position
 2: Initialize each Tree position
 3: **while** termination condition not satisfied:
 4:     $FP_{current}$ choose another one random FP, $FP_1$.
 5:     Get position of FP; $X_1 = FP_1$ position.
 6:     Calculate new position using Equation (3); $X_{new} = X_1 + RandomLevy() * 0.1$.
 7:     Update new position
 8:     Calculate distance between FP and nearest Tree; $d_{tree} = d(X_{new} - X_{tree})$.
 9:     **if** $d_{tree} \leq 3$ ($T_{nearest}$ found):
10:         **if** FP carry poll of $T_{poll} \neq T_{nearest}$:
11:             Pollination.
12:         **end if**
13:         FP carry poll of $T_{nearest}$; $T_{poll} = T_{nearest}$
14:     **end if**
15: **end while**

---

### 3.2.2. Individual Random Search

Individual random search refers to an individual who conducts random investigations by themselves without referring to other FP location.

**Method 4:** Lévy Flight

Lévy flight is well known for solving diffuseness, scaling, and transmission problems related to optimization. According to numerous studies, researchers find that the Lévy technique is universal and many innovations have evolved to boost Lévy flight efficiency [27]. Lévy flight is essentially a random walk, with the arbitrary stride length drawn from the Lévy distribution which has infinite variance and infinite mean. According to Reynolds and Frye's research [28], the fruit flies influenced the Lévy flight style's intermittent free-scale search pattern or *Drosophila melanogaster* exploring their environment with a succession of straight flight routes interspersed by 90° sudden twists.

A random walk generates Lévy flights with a stride length drawn from the stable levy distribution, as shown in Algorithm 4. A simple power-law formula is then described using the Lévy probability distribution. Here, $0 < \beta \leq 2$ is the Index of Lévy distribution [29].

$$Lévy(s) \sim |S|^{-1-\beta} \tag{4}$$

---

**Algorithm 4** Individual Lévy Flight.

---

1:  Initialize each FP position
2:  Initialize each Tree position
3:  Set $\beta$.
4:  Set list sigma value; *list_sigma*.
5:  Set reach of step; *reach*.
6:  **while** termination condition not satisfied:
7:      **if** $0.5 < \beta < 1.95$:
8:          Calculate index; $i = (\beta/0.05) - 1$.
9:          Get sigma, $\sigma = list\_sigma[i]$.
10:         Choose random value; $r_1, r_2$.
11:         Calculate normal distribution 1; $nd_1 = \sqrt{log(r_1) * (-2)} * cos(2\pi r_2) * \sigma$
12:         Calculate normal distribution 2; $nd_2 = \sqrt{log(r_1) * (-2)} * cos(2\pi r_2)$
13:         Calculate random Lévy; $levy = nd_1/|nd_2|^{(1/\beta)}$
14:     **end if**
15:     Calculate new position; $X_{new} = levy * reach * 0.1$
16:     Update new position.
17:     Calculate distance between FP and nearest Tree; $d_{tree} = d(X_{new} - X_{tree})$.
18:     **if** $d_{tree} \leq 3$ ($T_{nearest}$ found):
19:         **if** FP carry poll of $T_{poll} \neq T_{nearest}$:
20:             Pollination.
21:         **end if**
22:         FP carry poll of $T_{nearest}$; $T_{poll} = T_{nearest}$
23:     **end if**
24: **end while**

---

### 3.2.3. Multi-Agent System

A multi-agent system (MAS) is a system consisting of multiple interacting computing components known as agents. MAS appears to be a natural metaphor for understanding and building various types of what we can call an artificial social system. The concept of MAS is not reliant on a single application domain, but it seems to be prevalent across various application domains [30]. MAS is commonly used to model self-organizing systems and emerging behavior but has not been applied much to random searches and the immediate solution of optimization problems in a generic way. Since not tied to the direct evaluation of fitness functions, the Pollination Problem also offers MAS the prospect of an application.

**Method 5:** Defender-Aggressor-Game (DAG)

We used a basic participation game inspired the Defender-Aggressor-Game (DAG) in which each player chooses two other players at random. Assume that the selected players are player A and player B, as shown in Figure 6. Everyone in this game seeks to position themselves so that their A (the player's "Defender") is always between them and their particular B (the player's "Aggressor"). Everyone in this game tries to place herself with A and B in the same concurrent manner. This simple rule maintains the stable dynamics and keeps all agents moving around randomly, with a low chance that the pattern stabilizes to a line-like arrangement of all players [31].

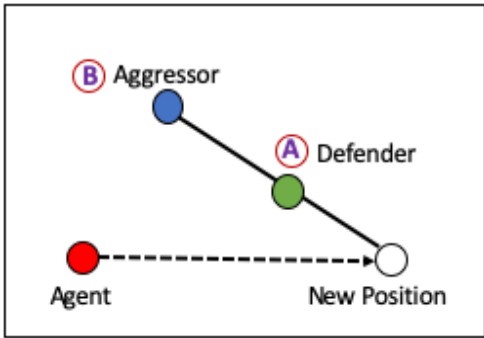

**Figure 6.** Rules for playing the Defender Aggressor Game.

It is stated in the Algorithm 5 that in order to obtain a new position, the position must be multiplied by the safety factor of 1.2. The safety factor is the parameter value that has been determined to move far enough behind the defender but not too far. A value above 1 but close to 1 is a common choice.

---

**Algorithm 5** DAG.

---

1: Initialize each FP position
2: Initialize each Tree position
3: **while** termination condition not satisfied:
4:     $FP_{current}$ choose two another random FP; $FP_1$ and $FP_2$.
5:     Repeat until $FP_1 \neq FP_2$.
6:     Get position of FP; $X_1 = FP_1$ position, $X_2 = FP_2$ position.
7:     $FP_1$ as aggressor, $FP_2$ as defender.
8:     $X_{new} = X_1 + (X_2 - X_1) * 1.2$
9:     Update new position.
10:     Calculate distance between FP and nearest Tree; $d_{tree} = d(X_{new} - X_{tree})$.
11:     **if** $d_{tree} \leq 3$ ($T_{nearest}$ found):
12:         **if** FP carry poll of $T_{poll} \neq T_{nearest}$:
13:             Pollination.
14:         **end if**
15:         FP carry poll of $T_{nearest}$; $T_{poll} = T_{nearest}$
16:     **end if**
17: **end while**

---

## 4. Results

The experiment results using the metaheuristic algorithm method from the three different techniques vary depending on the method's behavior. Figure 7 represents the results of these variations by showing the average tree pollination. The time is represented on the x-axis, which ranges from 0 to 20 min. The first minute represented on the graph indicates that the average pollination happened between 0:00 and 0:59. Figure 7a indicates that DAG pollinated more trees at the first minute with an average value of 3.8. Lévy comes in second with an average value of 2.35, followed by Idle-CSA 2.2, Idle-Cuckco, and Idle-Jaya with the same average score of 0.95.

All simulations demonstrate a substantial difference in results when using the swarm algorithm, i.e., Idle-Jaya, Idle-CSA, and Idle-Cuckoo, compared to searches that do not use swarm behavior approaches such as Lévy flight and DAG; the average percentage results. Swarm search algorithms such as Idle-Jaya, Idle-CSA, and Idle-Cuckoo need some time to pollinate more trees, but Lévy flight and DAG require less time to pollinate more trees. This simulation indicates that the swarm method in pollination here requires a longer period of time since these algorithms are meant to work together in the search. However, if using multi-agent system search and individual random search method, more trees are pollinated in a not too long time period since the range of each FP is different. The FP acts independently without the need to stay together. In other words, in using SI algorithms,

we could observe that the FP swarm all close to the same tree, where they can obtain all nectar but will not cross-pollinate before reaching another tree. Once the first FP arrives at the other tree, the tree becomes pollinated and the nectar will not be available to other FP anymore.

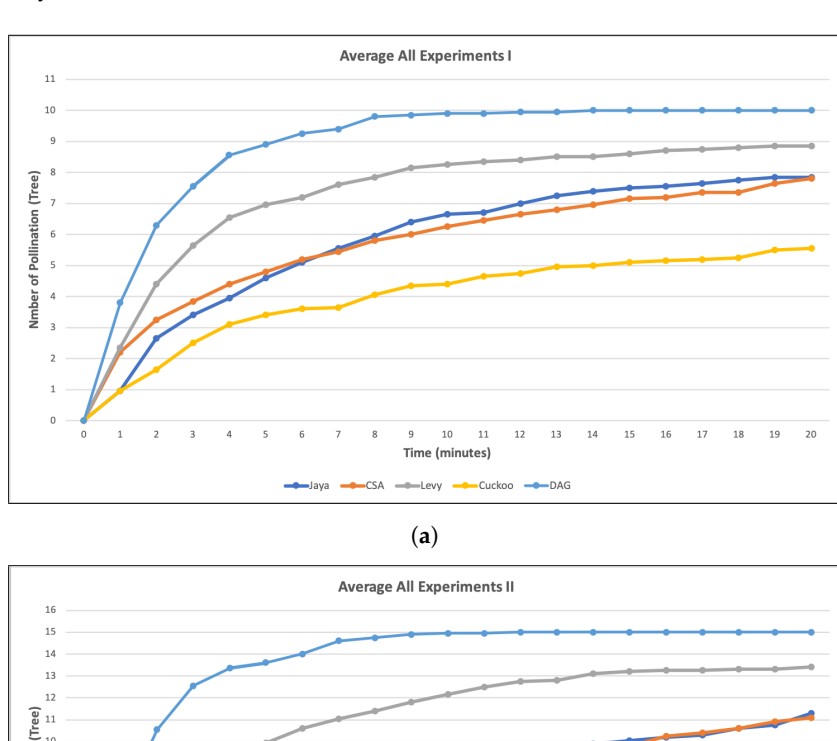

(**a**)

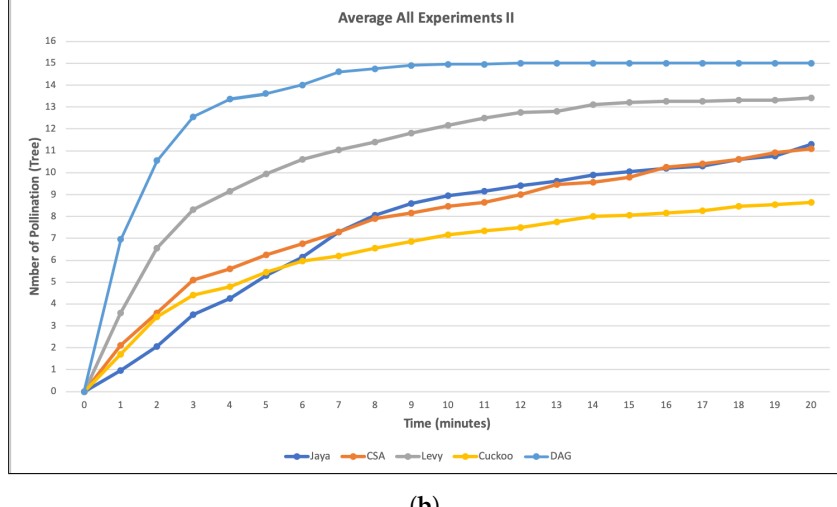

(**b**)

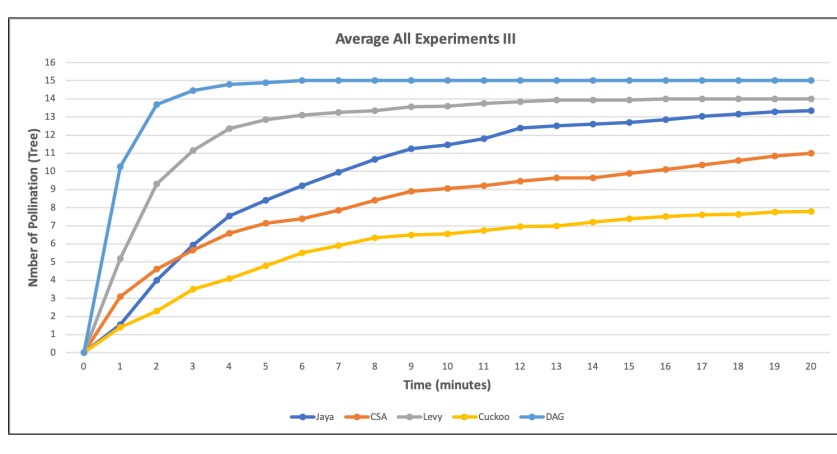

(**c**)

**Figure 7.** The average results from all simulation. (**a**) Experiment I. (**b**) Experiment II. (**c**) Experiment III.

Aside from the average tree pollination results shown above, this simulation also shows how much nectar FP collects during a tree search. The average results are shown in Table 2. The average amount of nectar collected from all simulations also shows that Lévy flight and DAG collect more nectar, even though the values of other Idle-SI are not significantly different from the results.

**Table 2.** The average nectar amount from all simulations.

| Algorithm | Experiment 1 | Experiment 2 | Experiment 3 |
| :---: | :---: | :---: | :---: |
| Idle-Jaya | 9.55 | 12.10 | 15.50 |
| Idle-CSA | 10.10 | 13.20 | 13.95 |
| Idle-Cuckoo | 9.00 | 15.10 | 10.40 |
| Levy Flight | 10.50 | 18.80 | 15.50 |
| DAG | 11.70 | 17.70 | 16.80 |

According to Morgan and the researchers of [32], the Gini index is the single best measure of inequality. The Gini index is a well-known concentration index established by Corrado Gini [33] more than a century ago to measure the level of inequality in income distribution and wealth distribution. The Gini index is used here to describe whether FP nectar intake is spread throughout the FP population in a impartial and balanced manner, contrary to a situation where few FP pick most of the nectar alone. The Gini index values for the FP are shown in Table 3. The Gini index from all experiments revealed that FP distributed nectar equally because zero represents the Gini index of perfect equality. All values are the same, whereas 1 expresses maximal inequality among FP. However, an advantage of DAG against other algorithms is notable here.

**Table 3.** The Gini Index average of the simulated representation of FP distribution nectar.

| Algorithm | Experiment 1 | Experiment 2 | Experiment 3 |
| :---: | :---: | :---: | :---: |
| Idle-Jaya | 0.53 | 0.47 | 0.63 |
| Idle-CSA | 0.47 | 0.39 | 0.55 |
| Idle-Cuckoo | 0.49 | 0.40 | 0.64 |
| Lévy Flight | 0.48 | 0.37 | 0.58 |
| DAG | 0.41 | 0.31 | 0.49 |

## 5. Discussion

After simulating the case of cocoa flower pollination by using various random search algorithms, the main finding is that Lévy Flight and DAG outperformed the selected SI search approaches, particularly Idle-Jaya, Idle-CSA, and Idle-Cuckoo Search (that also includes Lévy flight). Lévy flight and DAG could pollinate more trees than Idle-Jaya, Idle-CSA, and Idle-Cuckoo in the first several minutes of the simulation out of the three case simulations demonstrated. According to the paper [34], by comparing the search algorithm for neural architecture search (NAS), the evolutionary algorithm is better at handling optimization on NAS. Bn contrast, a random search may be faster but does not guarantee the best results in the case of the Pollination Problem.

The DAG method used here is not that different from general evolutionary methods. It can be related to a particular case of Differential Evolution (DE). The original DE procedure is summarized as follows: A simple differential mutation operation resulting from two different individuals chosen from the population to disturb a randomly selected individual as the base vector. Then generate progeny candidates and a one-to-one selection strategy to determine which individuals are still surviving [35]. The main point is that DE adds the difference vector between two other individuals to itself or a third randomly selected position for a new search candidate position. DAG essentially does the same by adding the difference vector of Defender and Aggressor to the Defender. Thus, it is a differential algorithm as well. DE is as known as a strong metaheuristic in many application cases

while being easy to use and implement. Based on the results, we can conclude that DAG outperforms other Idle-SI algorithms. DE is also a favoured algorithm for solving real-valued continuous optimization problems [36–38].

In addition to DAG, Lévy flight produces good results in second place after DAG. This result also supports the author's [39] proposal to include Lévy flight in the Jaya basic algorithm to improve exploration and exploitation capabilities during the search process. In this paper, Lévy flight is proposed to be incorporated into Jaya basic algorithm to facilitate the global search in the initial stages and local stages of the last investigation, increasing the algorithm's exploration and local optima avoidance capabilities. However, Idle-Cuckoo search, a derived SI algorithm from Cuckoo Search, already includes a Lévy flight and the results show the worst performance among all algorithms studied here.

What is the main benefit of such studies? After all, we cannot consider a head-to-head competition among algorithms as is common in other studies on the application of meta-heuristic algorithms, for example [40]. This is simply due to the lack of an immediately available fitness function. However, even when reducing such algorithms to their exploration component, it shows significant differences in the congruent goal of pollinating trees that are concomitant to nectar collection. We can observe that the three types of algorithms refer to cases of more general random search strategies:

- Individual strategies, such as Lévy flight, are such that each FP pursues its trajectory independent of the other FP. The case is most supported also by biological insights into insects flight patterns. We can testify that it is a good method but not the best quality method.
- The FP takes a reference to one or more other FP positions for deciding on the next move. This is basically implemented in all Swarm Intelligence algorithms. However, we can see that there is neither a significant advantage concerning nectar intake nor any gain in the number of pollinated trees. From a biological point of view, it also rests on the assumption that insects can recognize their species among other objects in the environment.
- A differential approach in which the FP decides the next step based on a reference to the offset between two objects in the environment. Our analysis clearly shows the advantage of this strategy: pollinating the most significant number of trees and including the fairest distribution of nectar within the population. Moreover, the pun is on "different objects" and not necessarily other FP to take as reference. It means that the same method might work as well, e.g., other insect species as reference points. This method is subject to further investigations.

Practically, the promotion of differential strategies can promote pollination, which would require related farm experiments. The other impact is on the study of related optimization problems from this class of deferred fitness problems. In fact, we can find numerous problems that have been hard to approach so far: the evolution of parasitism as an example from biology. Moreover, there are congruent constraints in the food supply chain, for example, producing farm goods to arrive in a new state at some consumer site.

## 6. Conclusions

Here, we studied the problem of Pollination optimization that came out to be a concurrent optimization problem with a deferred fitness evaluation. FP and trees act together in a seamless but also contingent way to achieve this objective.

We investigated three different random search methods: fitness-free versions of Swarm Intelligence Algorithm, i.e., Idle-Jaya, Idle-CSA, and Idle-Cuckoo Search; the Individual Random Search method, i.e., Lévy flight; and finally, the Multi-Agent Systems search method, i.e., Defender-Aggressor-Game (DAG). Those methods have been compared for simulating cocoa flower pollination.

We chose cocoa pollination as the simulation case because the flowers of the cocoa plant are unique from other plants. They are small (maximum diameter of 3 cm), allowing only small insects to pollinate them. Cross-pollination is the only method to ensure

successful fertilization because the self-pollination of unsuitable varieties will not result in successful fertilization.

From the results of this study, we can observe the differences among random search strategies. Concerning the main objective and the ratio of pollinated trees, there is an apparent gain from the differential method DAG. There are no significant differences relative to nectar intake and distribution, while there is a better value tendency of DAG. Generally, we can conclude that there is no best method at all points. However, we can refer to the results of this simulation for multi-agent and random searches, which may be more suitable for cocoa pollination in the real world. The simulation approach is also expected to assist a farmer when it comes to the inadequate pollination of cocoa flowers. This can improve the cocoa plant's productivity, as the related setup and experiments can also be performed by farmers using the same simulation software to learn about the influence of the various factors in pollination. By utilizing this simulation, farmers can use this simulation to manage or design their tree or plant placement and the order of potential nests or breeding site for pollinator nests.

**Author Contributions:** Conceptualization, W.A.S. and M.K.; methodology, M.K.; software, M.K. and R.M.P.; validation, W.A.S. and M.K.; formal analysis, W.A.S. and M.K.; investigation, W.A.S. and M.K.; resources, W.A.S. and M.K.; data curation, W.A.S. and M.K.; writing—original draft preparation, W.A.S.; writing—review and editing, M.K.; visualization, W.A.S.; supervision, M.K. All authors have read and agreed to the published version of the manuscript.

**Funding:** This research received no external funding.

**Data Availability Statement:** No new data were created or analyzed in this study. Data sharing is not applicable to this article.

**Acknowledgments:** The authors of this article would like to thank the Kyushu Institute of Technology for their financial and educational support.

**Conflicts of Interest:** The authors declare that they have no known competing financial interests or personal relationships that could have appeared to influence the work reported in this paper.

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
