# Peer review of "Behavior Selection Metaheuristic Search Algorithm for the Pollination Optimization: A Simulation Case of Cocoa Flowers"

_algorithms, doi:10.3390/a14080230_

Round 1
Reviewer 1 Report
The authors propose using modified existing metaheuristics to simulate the flower pollination problem. Below are my comments:
- The paper contains some grammatical errors and should, therefore, be proofread.
- The paper should have at least some related work mentioned.
- It would be more appropriate if Table 1 were placed in the experimental part of the papers since its information is more relevant.
- What does MH in Figure stand for?
- “We use the Idle-Jaya formula in the pseudo-code above” – in the paper, the pseudo-code is located below the text.
- What if the distance to FP1_pos and FP2_pos is the same in Algorithm 1?
- The pseudo-codes have many ambiguities, which make them hard to understand. For example, why does not Algorithm 1 update the amount of nectar? And what is fence_radius in Algorithm 2?
- Figure 5 should also have marked which is player A and which is player B.
- It is not very clear how exactly this can be applied to the real-world problem since no outside factors are considered that farmers could affect. How exactly can farmers use the obtained knowledge in pollination?
Reviewer 2 Report
The paper investigates the behavior of several metaheuristic optimizers to the problem of simulated cocoa flowers pollination. The search algorithms, which include swarm intelligence, random search and multi-agent approaches are applied as exploration techniques. Although this study presents an interesting application of the well-known methods, there are several significant problems which should be solved. First of all, the experimental setup is not described enough: Table 1 provides only several basic parameters, and the text does not provide full explanation to the following questions:
-What was the size of the search space? Were the particles (FPs) kept within them, or they were able to travel anywhere?
-What was the dimension of the search space, 2D or 3D? Did FPS need to fly up to the tree to a certain height?
-What was the range below which the pollination occurred, i.e. distance between particular FP and tree? In other words, what were the pollination conditions?
-If the simulation was performed for 20 minutes, what was the time step? In other words, how many movements each FP was allowed to make until the end of the simulation?
-How were the initial positions of FPs generated? What kind of distribution (uniform, normal, etc.) was used and what were the parameters?
-How big was the ratio of the area around a tree with respect to the total size of the search space? For example, if the distance to the tree should be less than 0.1m, and the search space was 100x100 meters, then only 10 trees would make the pollination very difficult.
-How were the trees located in the search space? On a grid, randomly, did their position change in different simulations?
-How much nectar is collected by an FP during one iteration with a tree? Did every tree have one virtual flower or many?
-Why in Figure 6 the number of pollination does not start from 0, and why did DAG have more pollinated trees then other methods already at 0 minutes?
-Why was the classical Particle Swarm Optimization (PSO) not applied here? It seems as the first and most suitable analogy, as it models biological species searching for food/water.
Considering the above questions, in the current form the performed study does not provide sufficient understanding of the experimental part, so it cannot be reproduced and therefore requires major revisions.
Some minor notes: please edit the equations in Algorithms 1-5, do not use sqrt, abs, etc. and use mathematical notations instead.
Line 289: The Gini index vakues for -> values
Round 2
Reviewer 1 Report
The authors have addressed all the comments. Therefore the paper can be accepted.